

# Evaluating the imazethapyr herbicide mediated regulation of phenol and glutathione metabolism and antioxidant activity in lentil seedlings

Rajeev Kumar[1], V. Visha Kumari[2], Ranjit Singh Gujjar[3], Mala Kumari[4], Sanjay Kumar Goswami[5], Jhuma Datta[6], Srikumar Pal[7], Sudhir Kumar Jha[8], Ashok Kumar[9], Ashwini Dutt Pathak[3], Milan Skalicky[10], Manzer H. Siddiqui[11] and Akbar Hossain[12]

[1] Division of Plant Physiology & Biochemistry, Indian Institute of Sugarcane Research, Lucknow, Uttar Pradesh, India
[2] Agronomy, Central Research Institute for Dryland Agriculture, Hyderabad, Telangana, India
[3] Crop Improvement, Indian Institute of Sugarcane Research, Lucknow, Uttar Pradesh, India
[4] Integral Institute of Agriculture Science and Technology, Integral University, Lucknow, Uttar Pradesh, India
[5] Crop Protection, Indian Institute of Sugarcane Research, Lucknow, Uttar Pradash, India
[6] Department of Agricultural Biochemistry, Bidhan Chandra Krishi Viswavidyalaya, Mohanpur, West Bengal, India
[7] Agricultural Biochemistry, Bidhan Chandra Krishi Viswavidyalaya, Mohanpur, West Bengal, India
[8] Division of Plant Biotechnology, Indian Institute of Pulses Research, Kanpur, Uttar Pradesh, India
[9] Division of Biochemistry, Indian Agricultural Research Institute, New Delhi, India
[10] Department of Botany and Plant Physiology, Faculty of Agrobiology, Food, and Natural Resources, Czech University of Life Sciences Prague, Prague, Czechia
[11] Department of Botany and Microbiology, College of Science, King Saud University, Riyadh, Saudi Arabia
[12] Soil Science, Bangladesh Wheat and Maize Research Institute, Dinajpur, Bangladesh

Corresponding authors
V. Visha Kumari,
visha.venugopal@gmail.com
Akbar Hossain,
akbarhossainwrc@gmail.com

## ABSTRACT

The imidazolinone group of herbicides generally work for controlling weeds by limiting the synthesis of the aceto-hydroxy-acid enzyme, which is linked to the biosynthesis of branched-chain amino acids in plant cells. The herbicide imazethapyr is from the class and the active ingredient of this herbicide is the same as other herbicides Contour, Hammer, Overtop, Passport, Pivot, Pursuit, Pursuit Plus, and Resolve. It is commonly used for controlling weeds in soybeans, alfalfa hay, corn, rice, peanuts, *etc.* Generally, the herbicide imazethapyr is safe and non-toxic for target crops and environmentally friendly when it is used at low concentration levels. Even though crops are extremely susceptible to herbicide treatment at the seedling stage, there have been no observations of its higher dose on lentils (*Lens culinaris* Medik.) at that stage. The current study reports the consequence of imazethapyr treatment on phenolic acid and flavonoid contents along with the antioxidant activity of the phenolic extract. Imazethapyr treatment significantly increased the activities of several antioxidant enzymes, including phenylalanine ammonia lyase (PAL), phenol oxidase (POD), glutathione reductase (GR), and glutathione-s-transferase (GST), in lentil seedlings at doses of 0 RFD, 0.5 RFD, 1 RFD, 1.25 RFD, 1.5 RFD, and 2 RFD. Application of imazethapyr resulted in the 3.2 to 26.31 and 4.57–27.85% increase in

mean phenolic acid and flavonoid content, respectively, over control. However, the consequent fold increase in mean antioxidant activity under 2, 2-diphenylpicrylhdrazyl (DPPH) and ferric reducing antioxidant power (FRAP) assay system was in the range of 1.17–1.85 and 1.47–2.03%. Mean PAL and POD activities increased by 1.63 to 3.66 and 1.71 to 3.35-fold, respectively, in agreement with the rise in phenolic compounds, indicating that these enzyme's activities were modulated in response to herbicide treatment. Following herbicide treatments, the mean thiol content also increased significantly in corroboration with the enhancement in GR activity in a dose-dependent approach. A similar increase in GST activity was also observed with increasing herbicide dose.

# INTRODUCTION

Lentil (*Lens culinaris* Medik.), is an edible cool-season legume, which contains ample amounts of high-quality protein. It also possesses carbohydrates, micronutrients, vitamins, phenolics, and flavonoids (*Khazaei et al., 2019*). Due to an immense weed invasion during the early crop growth cycle, the growth and development of the lentil plant is severely impeded which lowers yield and its contributing qualities (*Jaswal & Menon, 2020*). Based on the environmental factors, weed diversity, and density, the losses incurred are in the range of 20–80% (*Balech et al., 2022*). Proper management practices such as sowing method and time, cover crops, crop rotations, and varietal selection are customarily used to retard the growth and biomass of the weed and improve the lentil yield. However, these approaches are insufficient to curb weed interference effectively (*Pala, Mennan & Demir, 2018*). Compared to other methods, the effectiveness of herbicide treatment as of yet appears to be very high. It also acts quickly and requires little financial investment to control weeds (*Singh & Singh, 2017*). Imazethapyr (IM) application to manage the broad spectrum of weed flora has been advised as a means of reducing and controlling the weed threat in pulse crops (*Duary, Dash & Teja, 2016*). Imazethapyr belongs to the imidazolinone family and is applied as a post-emergence herbicide; a characteristic feature of this class includes its effectiveness at a lower dose, larger selectivity to various crops and lesser mammalian toxicity (*Tranel & Wright, 2002*). The selectivity of the imazethapyr is unique to control the multiple weed class at a lower rate of application (*Presotto et al., 2012*). Usually, imazethapyr application is found to be harmless to target crops and ecofriendly, when applied in lower amounts (*Hoseiny-Rad & Aivazi, 2020*).

Imazethapyr-mediated consequences on plant growth are primarily exerted through inhibition of the rate-limiting enzyme involved in branched-chain amino acids (BCAAs) synthesis (valine, leucine, and isoleucine) *i.e.*, acetolactate synthase (ALS) (*Qian et al., 2015*). Secondary consequences such as disruption in protein synthesis and cell division are associated with Imazethapyr-induced ALS inhibition. In addition to these issues, imazethapyr builds up in the plant's meristematic areas after being applied topically, which slows down the growth and development of the plant (*Hoseiny-Rad & Aivazi, 2020*). Even

though, ALS inhibition affects nitrogen metabolism through the reduction in protein synthesis rate as a consequence of a transitory halt in BCAAs synthesis (*Zabalza et al., 2006*). The plant's sessile nature restricts it from facing unavoidable environmental stresses, like drought, heat, freezing, high salt, xenobiotics contaminants including herbicide, *etc.*, which are the major abiotic factors that impact the productivity of the crop (*Kumar et al., 2023*). This results in substantial economic loss further raises the nutritional security issue (*Nianiou-Obeidat et al., 2017*). To survive under such stressful conditions plants have developed a potential defense mechanism that combats these stress eventualities (*Proietti et al., 2019*). The adaptive response mechanism of plants against several abiotic stresses implicates a complex system that is regulated through the coordinated action of multiple signalling compounds. These compounds include reactive carbonyl species (RCS), reactive oxygen species (ROS), phytohormones and reactive nitrogen species (*Fancy, Bahlmann & Loake, 2017*). ROS generated in plants during their exposure to herbicides creates a stress condition for plants (*Hassan & NematAlla, 2005*; *NematAlla, Hassan & El-Bastawisy, 2008*). Antioxidants are a crucial component of the plant defence mechanism that protects plants from oxidative stress brought on by herbicidal stress. The coordinated action of both enzymatic and non-enzymatic antioxidants in the plant system, produced upon exposure, controls ROS elimination (*Grewal et al., 2022*). Phenol and glutathione and their associated metabolism are excellent mechanisms to turn down the stress impact caused by the herbicide and impart tolerance against them. Regulation of phenol metabolism is primarily monitored through the phenylalanine ammonia-lyase (PAL) action and activity, the key enzyme involved at the entry valve of phenol metabolism (*Kong, 2015*). Furthermore, these synthesized phenol molecules oxidized through the coordinated action of oxidative enzymes such as polyphenol oxidase (PPO) and peroxidase (POD) (*Singh et al., 2022*). Besides, glutathione metabolism-associated enzymes viz. glutathione-s-transferase (GST) and glutathione reductase (GR) are also important players in the detoxification of ROS generated through herbicidal-induced oxidative stress (*Tseng, Ou & Wang, 2013*). Identification of herbicide dose and its associated tolerance mechanism at the early phase of the crop cycle in lentil would have potential use to recommend in lentil cultivation practice for control of weeds without compromising the plant growth and yield. Taking all these into consideration this present study was conducted to assess the dose-dependent response of imazethapyr on phenol and glutathione metabolism and its associated antioxidant potential in lentil seedlings at different sampling hours after its application. Furthermore, the current study will decipher the phenols and glutathione-associated tolerance mechanism against imazethapyr in lentils at the seedling stage.

## MATERIALS AND METHODS

### Experimental setup/establishment of settlings

The present investigation was carried out in a pot culture-based experiment under controlled conditions (net house) at the division of Agricultural Biochemistry, Bidhan Chandra Krish Vishwavidalaya, Mohanpur, India. Before sowing, seeds of lentil (cv. *Moitree* (WBL 77)) were sterilized (surface sterilization) with 3% sodium hypochloride up

to 10 min, following the thorough washing using distilled water. Properly washed seed subjected to soaking (8 h), and transferred in the pots of 30 cm width × 30 cm top height. Following the transfer of soaked seeds, seedlings were uniformly thinned to 30 plants in each plot after 10 days. Further, pots were divided and allocated to six groups, control (1), while the rest of the pots (5) were used to implicate the five different imazethapyr (10 SL) doses as outlined below. T0: Untreated control; T1: imazethapyr @ 12.5 kg ai/ha (0.5 RFD, 0.5 multiples of Recommended Field Dose); T2: imazethapyr @ 25.0 kg ai/ha (RFD, Recommended Field Dose); T3: imazethapyr @ 31.25 kg ai/ha (1.25 RFD, 1.25 multiples of the Recommended Field Dose); T4: imazethapyr @ 37.50 kg ai/ha (1.50 RFD, 1.50 multiples of the Recommended Field Dose); T5: imazethapyr @ 50.00 kg ai/ha (2.0 RFD, 2.0 multiples of Recommended Field Dose). Depending on the surface area of the pot and amount of soil on a per hectare basis, the required amount of imazethapyr was calculated for each herbicide dose, and each herbicide dose (amount) was then solubilized with the adequate quantity of water before being sprayed using a mechanical sprayer (knapsack) in a cross-wise direction. Each treatment was replicated in quadruplicate in a completely randomized block design. Collection of lentils shoot just before the application of imazethapyr (0 h) and subsequently at regular intervals of 30 h up to 120 h after treatment (HAT) was performed with proper care. The collected shoots were rinsed with a copious amount of water to avoid the soil particles and dried using tissue paper.

## Analysis of phenylalanine ammonia-lyase

The enzyme extract was prepared by grinding 1 g of fresh tissue (whole seedlings) in 0.1 M phosphate buffer (10 ml pH 7.5). Additionally, 2% polyvinylpyrrolidone and triton-x (0.25%) were used to prepare the extraction buffer. The extraction of plant samples (seedlings) was performed in pestle and mortar (pre-chilled). The homogenate obtained through maceration was centrifuged at 10,000 rpm for 30 min at 4 °C, and the obtained supernatant was used to conduct the enzyme assay. The extracted enzyme source was kept in an ice bath, before the enzyme assay set-up. Estimation of PAL activity was performed by mixing 0.1 M Tris-HCl pH 8.8 (1.9 ml), 1 ml substrate (0.01 M phenylalanine) and freshly prepared enzyme extract (chilled) (0.1 ml). Monitoring of change in the absorbance (ΔA) performed at a regular interval of 5 min up to 30 min at 270 nm. Further, estimation of the activity was carried out using the method of *Burrell & Rees (1974)* with required modifications. The specific activity of the enzyme was calculated by using the standard curve (*trans-cinnamic* acid). The specific activity of PAL against herbicide (imazethapyr dose) was expressed in µmol *trans*-cinnamic acid produced $h^{-1}$ $mg^{-1}$ of protein.

## Analysis of phenol oxidase

Estimation of phenol oxidase was performed using the method of *Shannon (1966)* with slight modifications. The reaction mixture for enzyme assay contains potassium phosphate buffer pH 7.5 (2.65 ml), methanol dissolved guaiacol (4%) (0.15 ml), $H_2O_2$ (1%) (0.15 ml) and 0.05 ml enzyme source. Furthermore, mixing of the reaction constituent was done (bottom-top shaking approach) in a fraction of a second. The observation of absorbance change (ΔA) was monitored at a regular interval of 30 s to 3 min at 470 nm. Calculation of

phenol oxidase was accomplished by using the absorption coefficient of the tetra guaiacol at 470 nm (26.6 mM$^{-1}$cm$^{-1}$). The specific peroxidase activity expressed in μmol of guaiacol oxidized min$^{-1}$mg$^{-1}$ of protein.

## Analysis of total phenol and flavonoid contents

The total phenolic acid content of lentil shoots was estimated by using Folin-Ciocalteau Reagent (FCR) following the method of Vinson (*Vinson et al., 1998*) with required modification. Briefly, the dried powdered (0.1 g) lentil shoot sample was mixed in 15 ml 1.2 N HCl prepared in aqueous methanol (50%). Furthermore, the methanolic extract was subjected to heat (2 h at 90 °C). Extracted material following the cooling (kept at room temperature) was subsequently centrifuged for 30 min at 10,000 rpm. Furthermore, the supernatant in the volumetric flask and let it evaporate to become dry, finally, the volume was made to 25 ml with double distilled water. For the evaluation of phenol content, a particular aliquot volume was taken, further dilution (3 ml) with double distilled water was carried out in a test tube, and finally, the addition of 0.5 ml FCR in the diluted methanolic extract was accomplished. Subsequently, sodium carbonate (10%, 2 ml) was added after 5 min of FCR addition. Then, it was placed in a water bath for 7 min at 65–70 °C. The reaction was stand allowed to cool (room temperature) and the absorbance of the solution was taken at 650 nm. A standard curve using 8 different phenolic acids, namely gallic acid, chlorogenic acid, hydroxybenzoic acid, p-protocatechuic acid, vanillic acid, caffeic acid, p-coumaric acid, and ferulic acid was prepared as per the method ascribed by *Alla & Younis (1995)* with slight modification. Each of the phenolic acids represents its respective concentration (5 ppm), which corresponds to the final concentration of total phenolic acids at 40 ppm. The standard curve was used for the calculation of total phenolic acid in lentil shoots (seedlings) and expressed in mg phenolic acid g$^{-1}$.

In addition, a standard of flavonoids was prepared following the Folin-Ciocalteau assay using the mixture of seven different flavonoids comprising apigenin, myricetin, quercetin, genistein, catechin, kaempferol, and diadzein. All but two of the flavonoids were at a concentration of 5 ppm each, while apigenin and myricetin were added at a concentration of 2.5 ppm each, which corresponds to a total flavonoid concentration of 30 ppm. Thus, the flavonoid content of lentil shoots was measured based on the standard curve of flavonoids and expressed as mg flavonoid g$^{-1}$ sample.

## Measurement of antioxidant activity

Because of the difficulty in the measurement of phenolic composition with their antioxidant role in plant tissue, antioxidant activity may conveniently be signified as a measure of the quality of phenol. The antioxidant action of a substance is a measure of the capability to transfer an electron either in the form of hydrogen atom transfer (HAT) or a single electron transfer (SET). In the current study, total phenolic extracts were used to determine the antioxidant activity of lentil seedlings by using DPPH and FRAP assay, which uses antioxidant mechanisms involving HAT and SET, respectively.

## DPPH assay

DPPH is a neutral radical, which is extensively used to measure antioxidant activity in clinical studies. Hydrogen atom transfer from the substrate reduces the DPPH radical, which is accompanied by a decrease in the intensity of colour as well as absorbance of the solution. The DPPH assay was executed using the method adopted by *Braca et al. (2001)*. The reaction mixture consisting of 150 µL of the aliquot of total phenol extract and 2,850 µL of the DPPH solution (0.004%) was taken and properly mixed by hand-shaking. Furthermore, it was kept at normal temperature for 30 min in a dark place. The absorbance of the solution was taken at 517 nm, for blank distilled water was used along with DPPH. The standard curve was prepared by using 150 µL of each of the different concentrations of Trolox and 2,850 µL of 0.004% DPPH solution. The antioxidant activity, also termed as DPPH-generated radical scavenging capacity is expressed as a milligram of trolox equivalent per gram of fresh weight (mg TE $g^{-1}$ FW).

## FRAP assay

FRAP system of the antioxidant assay has relied on the capability of phenol extract to reduce Fe (III), which was measured according to the Benzie and Strain (*Benzie & Strain, 1996*) method with slight modification. Changes in the absorbance owing to the blue-coloured compound Fe(II)-tripyridyltriazine formation from the colourless parent compound containing the oxidized form of Fe (III), were assayed in the FRAP system, and the presence of the unknown concentration in phenolic extract was monitored. Preparation of the FRAP reagent was accomplished through the mixing of acetate buffer (0.1 M, pH 3.6), 2,4, 6-tri (2-pyridyl)-s-triazine (TPTZ) (10 mM) and ferric chloride (20 mM) in 10:1:1 (v/v/v) proportions. The reaction mixture comprising 2,850 µL reagent (FRAP) and 150 µL sample aliquot was kept at normal temperature (30 min). Thereafter, absorbance was taken at 593 nm. Different concentration of Trolox was used for the standard curve preparation. Finally, results were presented as milligrams of trolox equivalent to per gram of fresh weight (mg TE $g^{-1}$FW).

## Analysis of enzyme activity of glutathione-s-transferase and glutathione reductase

To assay the enzyme activity of both GST and GR, the enzymatic extract was prepared by macerating the fresh tissue (1 g whole seedlings) in 10 ml of phosphate buffer (0.1 M pH 7.5). Additionally, 7.5% PVP, 14 mM β-mercaptoethanol and 2 mM EDTA were added to the extraction buffer, and crushing of plant tissue (whole seedlings) was performed in a pre-chilled mortar pestle. The homogenate obtained was transferred into the centrifuge tube. Subsequent to extraction plant samples (seedlings) were subjected to centrifuge for 30 min at 10,000 rpm and 4 °C. The obtained supernatant was used to assay both GST and GR activities. The activity of glutathione reductase was assayed following the procedure of *Rao, Paliyath & Ormrod (1996)* with minor modification. The principle of this method relies upon the oxidation of assimilatory power (NADPH) through oxidized glutathione (GSSG). The reaction set up for the GR assay comprises Tris-HCl buffer (2.3 ml, pH 9.0), 0.1 ml GSSG (5.44 mM), 0.1 ml EDTA and 0.2 ml enzyme source. The reaction was commenced

by adding 0.2 ml of NADPH (0.2 mM). Subsequently, the absorbance of the solution was observed at 340 nm at a regular interval (1 min up to 5 min). Furthermore, the activity of GR was calculated by using the NADPH extinction coefficient (6.2 mM$^{-1}$ cm$^{-1}$) at 340 nm. The specific GR activity was presented in μmol of NADPH oxidized min$^{-1}$mg$^{-1}$ of protein. The reaction mixture in GST assay comprises 2.4 ml of potassium phosphate buffer (pH 6.5), 0.2 ml of GSH (5.0 mM), 0.2 ml of 1 mM CDNB and enzyme extract (0.2 ml). Subsequently, measurement of absorbance (OD) was taken at 340 nm at the interval of 1 min and the last reading was taken at 5 min. The specific activity of GST was calculated based on the CDNB extinction coefficient (9.6 mM$^{-1}$cm$^{-1}$) following the Ando methods (*Ando et al., 1988*), with required modification which is based on the conjugation of CDNB to GSH.

### Analysis of total thiol

The amount of total thiol in lentil seedlings was measured using the method of *Maas et al. (1987)* through minor modifications. The homogenization of the plant sample (0.5 g) was performed in 0.02 M EDTA. Extraction was followed by centrifugation of the homogenate at 10,000 rpm for 30 min at 4 °C. The obtained supernatant was used for the thiol estimation. To set up the reaction 0.5 ml of sample extract, 0.7 ml buffer (Tris-HCl, 0.2 M and pH 7.5), 0.3 ml DTNB (10 mM) and cold methanol (3.5 ml). Furthermore, it was incubated (15 min) at normal temperature and absorbance was recorded at 412 nm. The calculation of total thiol concentration was done using the standard curve of reduced glutathione prepared for different concentrations. The result was expressed as μmole of GSH g$^{-1}$ fresh tissue (μmole GSH g$^{-1}$FW). Statistical analysis was performed by using the SPSS Professional Statistics ver. 7.5 (SPSS Inc., Irvine, CA, USA). Calculation and graphical presentation of the data executed in M/S Excel software.

# RESULTS

### PAL and POD activity in lentil

Phenolic compound metabolism is chiefly regulated and controlled through the coordinated action of various enzymes involved in the synthesis as well as the breakdown of the compounds. Conversion of L-phenylalanine (L-Phe) to cinnamic acid (*trans*-CA) is catalyzed through phenylalanine ammonia lyase (PAL, EC 4.3.1.24) *via* non-oxidative deamination reactions. Peroxidase (POD) and polyphenol oxidase (PPO) also play a major role in phenolic compound metabolism.

Therefore, PAL and POD activity was evaluated in the current investigation to acquire more evidence on imazethapyr-induced modifications in phenol metabolism as a stress reaction. Figures 1A and 1B, respectively, show the results for PAL and POD activity as affected by the use of various imazethapyr treatments. The results show that mean PAL and POD activity irrespective of sampling hours increased significantly above control in response to increasing concentration of imazethapyr treatments. The increase in mean PAL and POD activity showed a 1.63–3.66 and 1.71–3.35-fold variation as compared to the control. Moreover, the mean PAL activity did not show any significant variation between 0 HBT and 30 HAT, but thereafter increased significantly throughout the

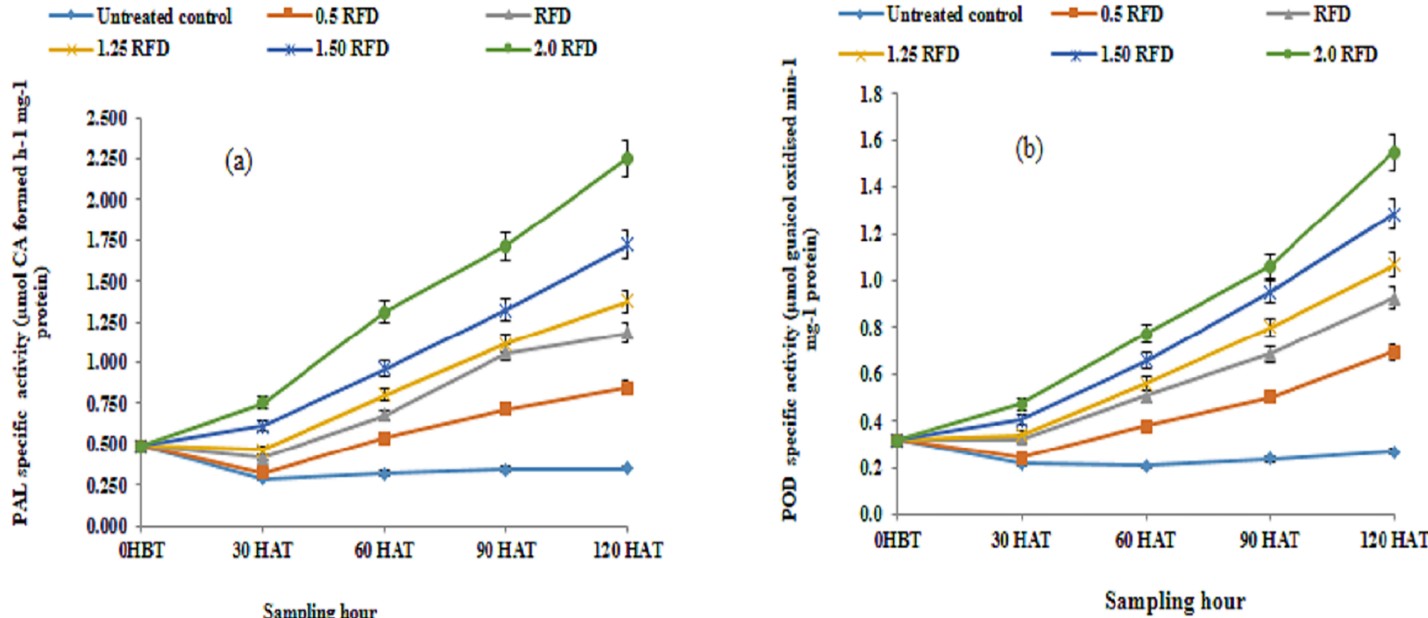

**Figure 1** **(A) Phenylalanine ammonia-lyase (PAL) and (B) phenol oxidase (POD) activity of imazethapyr treated seedlings at 0 HBT and 30, 60, 90 and 120 HAT under treatments of 0, 0.5, 1, 1.25, 1.5 and 2 RFD.** HBT, hours before treatment; HAT, hours after treatment; RFD, recommended field dose. The vertical error bar represents a standard error ($n = 4$).

experimental period. Regardless of treatments, PAL activity increased by 4.67-fold on 120 HAT as compared to 0 HBT. Contrary to PAL activity, significant differences in mean POD activity were observable and increased progressively throughout the experimental period. The enhancement in POD activity varied from 1.04–3.02 folds as compared to 0 HBT. Thus, both enzymatic activities are modulated by the treatment and sampling hours. In addition, the interaction between treatment (imazethapyr doses) and sampling hours was also significant. PAL, a marker of several kinds of abiotic stresses including herbicides, channels aromatic amino acids, chiefly phenylalanine to diverse phenolic compounds with equally diverse biological functions, which are related to ameliorating diverse environmental challenges. In the current study, a substantial increase in the mean PAL activity in response to an increased rate of the imazethapyr application and these effects are pronounced with the progression of the growth stage are noteworthy.

## Phenolic acid and flavonoid content in lentil

Following the application of imazethapyr at five different concentrations, phenolic acid and flavonoids were analyzed periodically before and after herbicide treatment. The results are summarized in Figs. 2A and 2B. The results obtained indicate that both mean phenolic acid and flavonoid content in lentil shoots increased significantly with the increasing application rate of imazethapyr. The mean phenolic acid over different sampling hours varied from 0.152–0.192 mg g$^{-1}$ FW, while the corresponding values for flavonoids were between 0.219 and 0.280 mg g$^{-1}$ FW. Hence, the application of imazethapyr resulted in the elevation of phenolic acid to the extent, which ranged from 3.2% to 26.31% over control, with maximum elevation recorded at 2 RFD and least at 0.5 RFD. A similar increase in

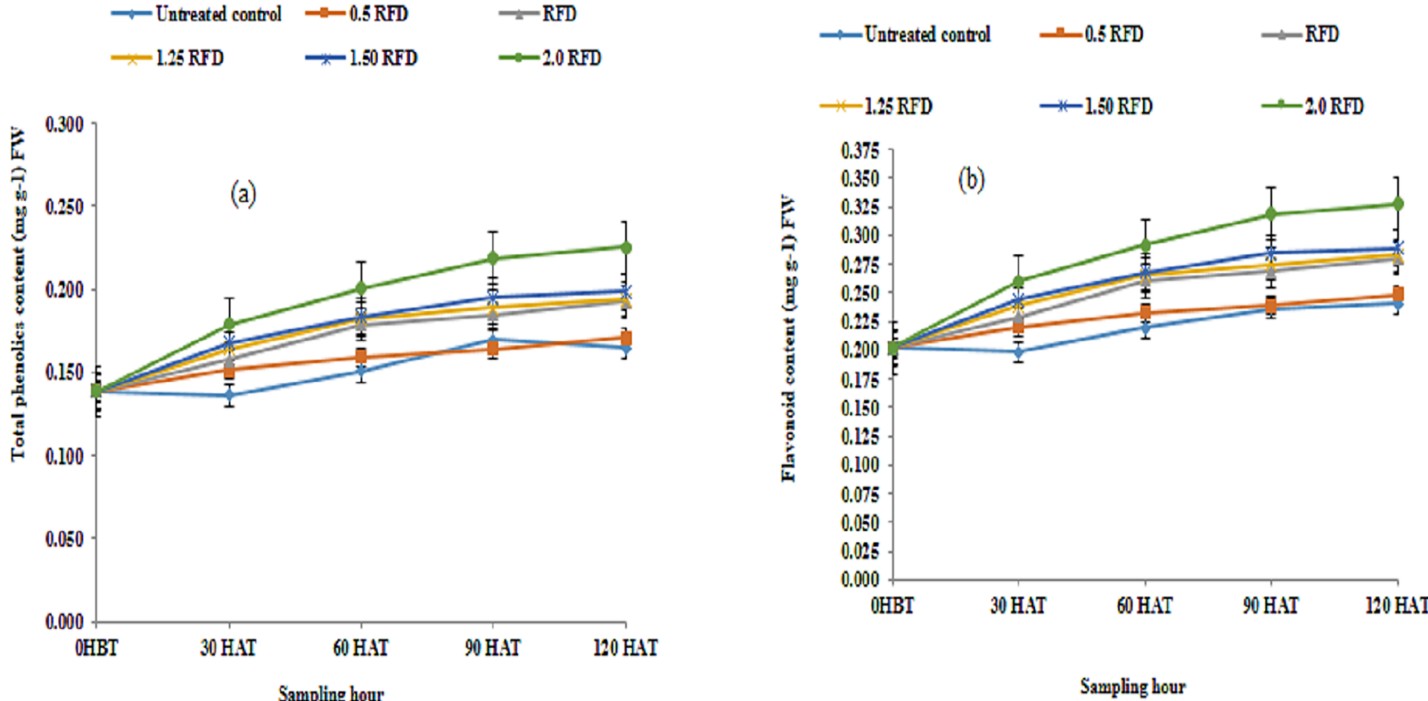

**Figure 2 (A) Total phenolic acid (mg g⁻¹) FW and (B) flavonoids content (mg g⁻¹) FW of imazethapyr treated seedlings at 0 HBT and 30, 60, 90, and 120 HAT under treatments of 0, 0.5, 1, 1.25, 1.5 and 2.0 RFD.** HBT, hours before treatment; HAT, hours after treatment; RFD, recommended field dose. The vertical error bar represents a standard error ($n = 4$).

flavonoid content, varying from 4.57–27.85%, was also observed depending on the herbicide dose. However, on the other side, the mean phenolic acid and flavonoid content over different treatments increased significantly with the progression of growth at early stages. During 120 h, the phenolic acid and flavonoid content increased over 0 HBT by 38.41 and 37.62%. The interaction outcome among herbicide dose and sampling hours also revealed significant involvement, indicating that phenolic acid and flavonoid content in lentil shoots changes depending on application rate as well as growth stage.

## Antioxidant activity in lentil

Diverse types of biomolecules known as antioxidants strongly resist the oxidant molecules produced by free radicals and oxidation driven by them, protecting the cellular biomolecules, even when present in minute quantities alongside other oxidizable substrates. Various biomolecules of diverse structural groups are recognized to act as an antioxidant, which includes phenols, tocopherols, ascorbic acid, glutathione, *etc.*

The shared central feature in these compounds lies in their ability to scavenge radical species. In the present study, the total phenolic extracts were analyzed for antioxidant activity using DPPH radical and FRAP, which exemplify a hydrogen atom transfer (HAT) and Single Electron Transfer (SET) mechanism of antioxidant reaction respectively. These mechanisms specify and exemplify the antioxidant reaction catalyzed by them.

The antioxidant activity in phenolic extract executed through DPPH and FRAP assay are summarized in Figs. 3A and 3B respectively. It is evident from Figs. 3A and 3B, that mean

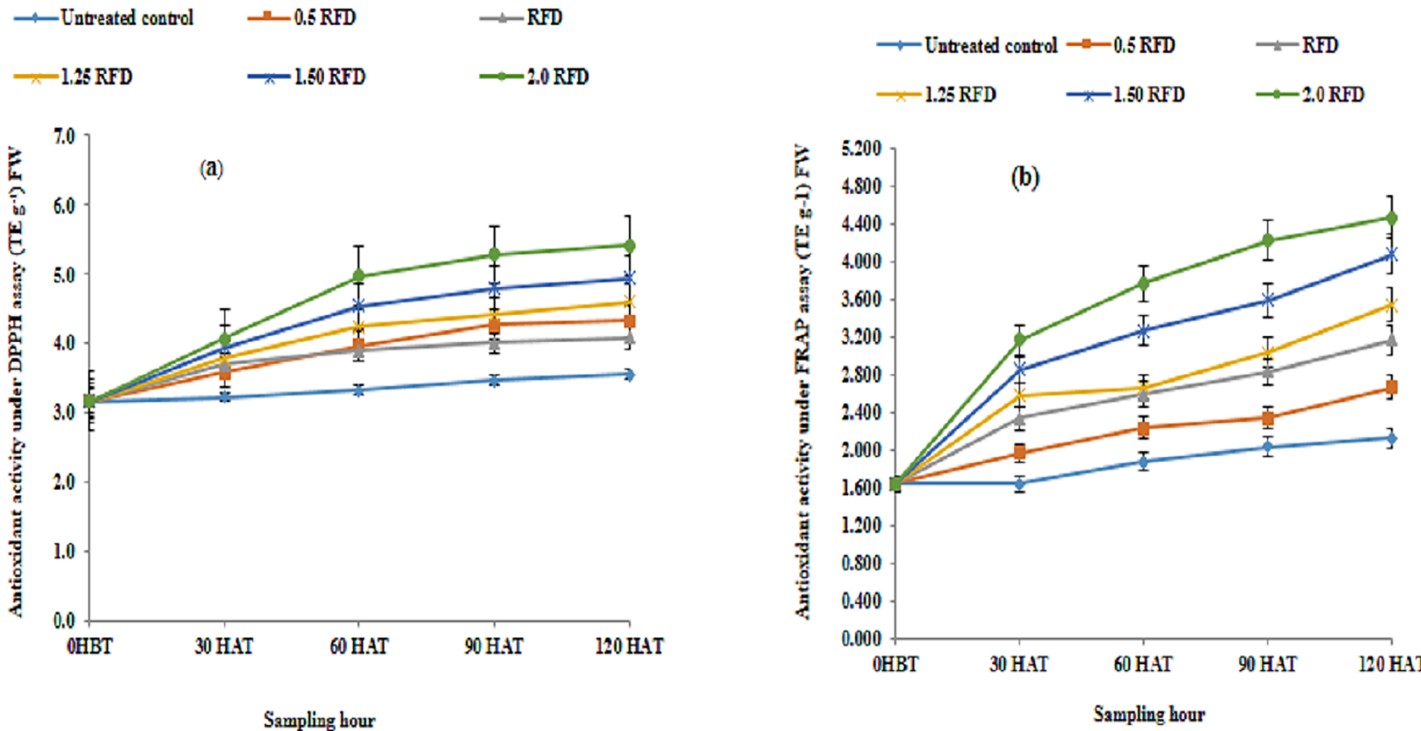

**Figure 3** **Total phenol extract (mg TE g$^{-1}$) FW in imazethapyr-treated seedlings using (A) DPPH and (B) FRAP assay at 0 HBT and 30, 60, 90, and 120 HAT under treatments of 0, 0.5, 1, 1.25, 1.5 and 2.0 RFD.** HBT, hours before treatment; HAT, hours after treatment; RFD, recommended field dose. The vertical error bar represents a standard error ($n = 4$).

antioxidant activity under both assay systems showed significant differences across different treatments and increased with increasing rates of application.

## GR and GST activities in lentil

The effect of imazethapyr on GR and GST activities is summarized in Figs. 4A and 4B, respectively. Both GR and GST activities enhanced substantially with the increased application rate of herbicide and progression of growth. The mean GR activity varied from 27.995–66.041 µmol of NADPH oxidized min$^{-1}$ mg$^{-1}$ protein, which represented an increase in GR activity ranging from 55.79 to 136.24% over the untreated control. However, the corresponding increases in GST activity were in the range of 61.30% to 157% over control. The mean GR activity showed a decline on 30 HAT and increased thereafter, while the mean GST activity increased progressively throughout the experimental period.

The GSH/GSSG proportion was higher with the pretilachlor-treated plant while least in metribuzin, GR and GST activities were stimulated more with pretilachlor than metribuzin in maize leaves. These herbicides, thus, induced oxidative stress differentially in maize, which is more, pronounced with metribuzin than with pretilachlor. Henceforth, it indicates the differential tolerance resulting from the enhancement in GSH content and stimulation of its associated enzyme activity. The elevation in the production of antioxidants such as phenol, and thiols and the prominent increase in the antioxidant

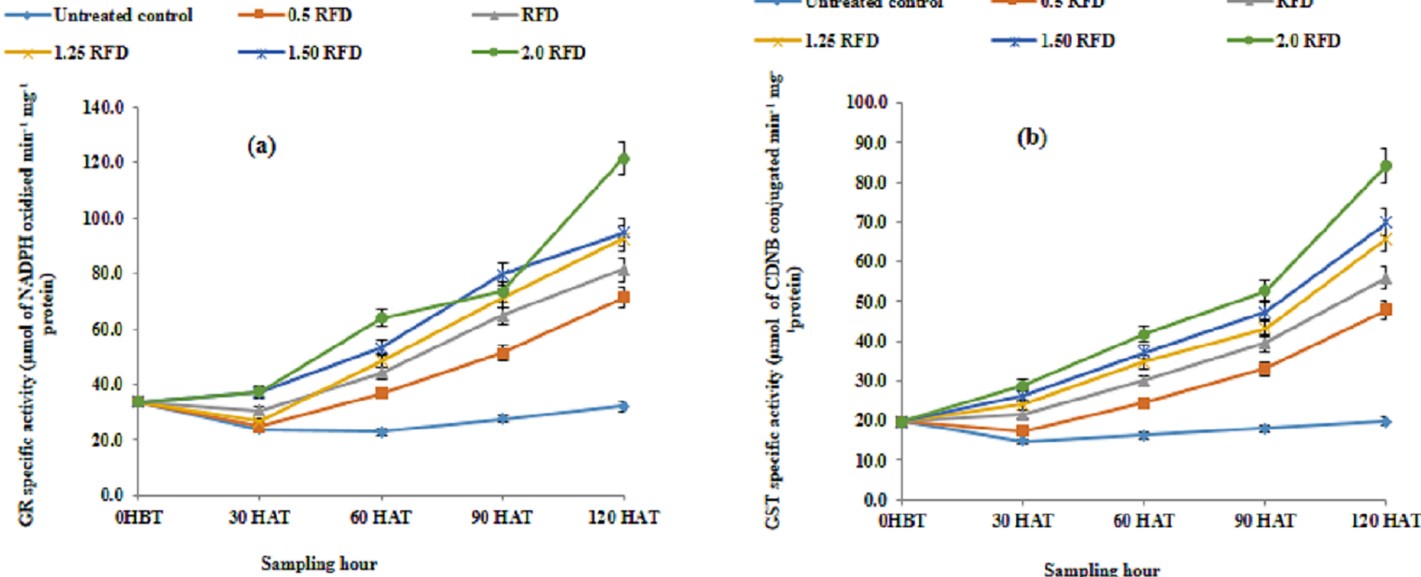

**Figure 4** **(A) Glutathione reductase (GR) and (B) glutathione-s-transferase (GST) activity in imazethapyr treated seedlings at 0 HBT and 30, 60, 90, and 120 HAT under treatments of 0, 0.5, 1, 1.25, 1.5 and 2 RFD.** HBT, hours before treatment; HAT, hours after treatment; RFD, recommended field dose. The vertical error bar represents a standard error ($n = 4$).

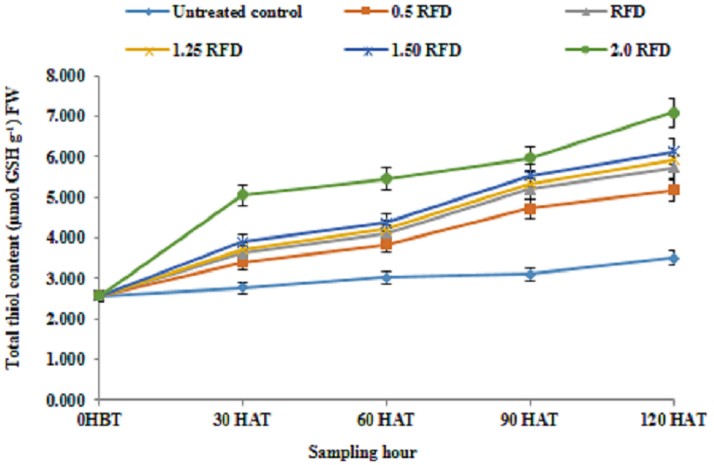

**Figure 5** **Total thiol content (μmol GSH/g FW) of imazethapyr-treated seedlings at 0 HBT, and 30, 60, 90, and 120 HAT under treatments of 0, 0.5, 1, 1.25, 1.5, and 2 RFD.** HBT, hours before treatment; HAT, hours after treatment; RFD, recommended field dose. The vertical error bar represents the standard error.

enzyme associated with these compounds like GR, GST, PAL and POD, appear to be the general strategies of the plant defense system to restrict the toxic peroxidation in plants.

## Thiol content in lentil

The results relating to total thiol content in lentil shoot following the application of imazethapyr at five different doses is depicted in Fig. 5, which revealed that the mean thiol content of lentil shoot showed significant differences depending on the treatment and

sampling hour. The mean thiol content regardless of sampling hours was recorded higher with 2 RFD (5.22 μmol GSH/g FW), and that decreased with decreasing rates of application. The thiol content was recorded lowest with 0.5 RFD (3.94 μmol GSH/g FW).

## DISCUSSION

The imidazolinone group of herbicides generally work for controlling weeds by limiting the synthesis of the aceto-hydroxy-acid enzyme, which is linked to the biosynthesis of branched-chain amino acids in plant cells (*Presotto et al., 2012*). The herbicide imazethapyr is from the class and the active ingredient of this herbicide same as other herbicides Contour, Hammer, Overtop, Passport, Pivot, Pursuit, Pursuit Plus, and Resolve. It is commonly used for controlling weeds in soybeans, alfalfa hay, corn, rice, peanuts, *etc*. Generally, the herbicide imazethapyr is safe and non-toxic for target crops and environmentally friendly when it is used at low concentration levels (*Qian et al., 2015*; *Duary, Dash & Teja, 2016*; *Hoseiny-Rad & Aivazi, 2020*). Although crops are extremely susceptible to herbicide treatment at the seedling stage, there have been no observations of its higher dose on lentil (*Lens culinaris* Medik.) at that stage (*Hanson & Thill, 2001*; *Grewal et al., 2022*).

### PAL and POD activity in lentil

Imazethapyr-induced dose-dependent, concurrent increase in the PAL and POD activity in the present investigation has been observed. This concomitant enhancement in activity indicates the possibility of a metabolic shift from primary to secondary to defend the plant against imazethapyr-generated free radicals. This shift in the present study is visualized through an increase in secondary metabolic compounds such as phenolic acid and flavonoid contents. It is likely to be concluded that the buildup of phenolic acid and flavonoid in lentil shoots with increasing herbicide dose cannot be explained solely based on enhanced PAL activity. Moreover, increased POD activity illustrates the availability of oxidizable substrates in the form of phenolic compounds upon increasing the herbicide dose. These increased substrate availabilities for the POD manifested the limiting role of PAL in phenol metabolism. However, enhanced POD activity may further lead to a more diverse class of compounds that enrich the plant defence armoury to face the herbicide-associated consequences. Therefore, PAL appears to be a key enzyme to regulate phenol accumulation in lentil seedling and POD acted in tandem. The enhanced activity of PAL further confirmed with increasing phenolic acid is reported in herbicide-treated seedlings of soybean and maize (*Alla & Younis, 1995*). Thus, imazethapyr-induced stimulation in the PAL activity complemented with increased phenol content in our study represents additional evidence for the effect of herbicides on phenol metabolism. Enhanced PAL and TAL (Tyrosine ammonia lyase) in response to imazethapyr in soybean root and shoot has been reported as a stress symptom (*Scarponi, Martinetti & Nemat Alla, 1996*). A large number of herbicides, including acifluorfen (*Kömives & Casida, 1982*), metolachlor (*Scarponi, Alla & Martinetti, 1992*), and alachlor (*Molin, Anderson & Porter, 1986*), also caused increased PAL activity, which is concerning for this relationship. Therefore, PAL appears to be a key enzyme to regulate phenol accumulation in lentil

seedling and POD acted in tandem. which is supported by Scarponi's report (*Scarponi, Alla & Martinetti, 1992*). Metribuzin induced an increase in the activities of peroxidases such as ascorbate peroxidase, guaiacol peroxidase, and the polyphenol oxidase observed in wheat (*Rajabi et al., 2012*). Moreover, *Islam et al. (2017)*, suggested the role of butachlor at a lower dose in the enhancement of GPX activity in rice (cv. ZJ 88), while at the higher dose, the activity of the enzyme was downregulated. Thus, our results do not align with the observation of Islam et al., which perhaps describes the substantial heterogeneity among the crop species and the difference in the nature of herbicides. Imazethapyr-induced enhanced expression of PAL isoforms in Arabidopsis root in a proteomics-based study has been observed (*Qian et al., 2015*). Biosynthesis of a large class of physiologically active secondary metabolites, phenylpropanoids such as flavonols, isoflavonoids, lignins and anthocyanins derived from phenylalanine catalyzed through the action of PAL (*Weisshaar & Jenkins, 1998*). In the present study imazethapyr mediated increased PAL and POD activity, consequent increase in phenolic acid and flavonoid contents in lentil seedlings indicates some sort of secondary metabolism-associated mechanism activated for quenching of free radical generated through this herbicide upon its exposure.

## Phenolic acid and flavonoid content in lentil

Phenolic compounds are widespread plant secondary compounds that play an important role during ecological imbalances (*Curir et al., 1990*). This class of biomolecules are involved in diverse processes like rhizogenesis, vitrification, redox reactions, and stress resistance (*Takahama & Oniki, 1992*). In the present investigation increasing phenolic acid and flavonoid content in lentil seedlings with increasing imazethapyr dose suggested the major role played by both of the metabolites in regulating the redox balance of the cell. The biological utility of the phenolic compounds is obtained through their involvement in the oxidation-reduction process (*Narwal, Kumar & Verma, 2016*). Enhanced activity of the POD at higher doses substantiates the active participation of phenolic compounds in the redox process by providing the substrate for their further oxidation. In this way, oxidative stress imposed through imazethapyr in lentil seedlings can be neutralized. Synthesis of the phenolic compounds in plants can be primarily regulated and influenced through various chemical stimuli induced due to an unfavourable environment that is perceived by the plants. Herbicide application leads to cellular stress and is associated with changes in the concentration of the phenolic compounds in plants. Increase in the level of these compounds were reported in response to some herbicide, while others showed declining effect (*Hoagland, 1990*). The current study documented noteworthy differences in the mean shoot (seedlings) phenolic acid and flavonoid content across different treatments. Moreover, imazethapyr stimulated the accumulation of phenol in lentil, which is further noticeable at a higher dose (2 RFD). Similarly, higher phenol accumulation was also reported in alachlor-treated soybean and maize (*Alla & Younis, 1995*), oat seedling treated with glyphosate (*Falco, Vilanova & Segura, 1989*), acifluorfen-treated spinach (*Kömives & Casida, 1982*) and alachlor sprayed sorghum (*Molin, Anderson & Porter, 1986*). Thus forth, Imazethapyr forms a prototypical representative in the expanding catalogue of herbicides that can moderate phenol metabolism as well as its accumulation in

lentil seedlings. Earlier studies indicate that herbicides trigger the rate of ROS formation (*Yin et al., 2020*). Additionally, increased phenol buildup in response to herbicides gives plants the ability to detoxify these species produced by free radicals. Basically, the plant's use of phenol in its detoxification process suggests a non-enzymatic form of reaction. Additionally, it offers the starting point for lignin biosynthesis. The level of phenolics, which is increased when alachlor is applied, controls the growth of soybean and maize seedlings, illuminating the influence of phenolics on growth behaviour when herbicides are applied (*Alla & Younis, 1995*). Imazethapyr is known to retard growth by inhibiting ALS leading to reduced synthesis of BCAAs and formation of protein. Although, it triggers the phenol accumulation rate in a dose-dependent manner. Thus, the dose-dependent accumulation of phenol in response to different imazethapyr treatments in the current study appears to form a component of the plant defense machinery, which eventually minimizes oxidative stress generated by the herbicide.

## Antioxidant activity in lentil

Phenolic compounds have been illustrated to modulate several biological processes that include antioxidant activity as well (*Kähkönen et al., 1999*). The edible as well as non-edible part of plants is the common source of these compounds (*Heim, Tagliaferro & Bobilya, 2002*). Earlier studies suggested the inherent capacity of plants gets activated upon herbicide exposure to defend against the generated consequences through their defense machinery (*Seneff, Swanson & Li, 2015*). In our study increased antioxidant activity in lentil seedlings upon imazethpyr application indicates the activation of these defense mechanisms through enhanced activity of PAL and POD and increased synthesis of phenolic acid and flavonoids. Numerous studies indicated a robust positive correlation between phenol concentration and associated antioxidant activity (*Kähkönen et al., 1999*). Herbicides that enhance phenol accumulation in plants, also lead to consequent stimulation of antioxidant activity (*Seneff, Swanson & Li, 2015*). Increased antioxidant activity in lentil seedlings in the present study at higher doses suggests the potential role of phenolic compounds in overcoming the herbicidal effects. Imazethapyr-induced phenolic levels exhibited higher antioxidant activity in mung bean and demonstrated a positive relation between both traits (*Namrata et al., 2020*). However, reports on high antioxidant activity exhibited under *in vitro* systems suggested that there is a meagre chance to combat ROS *in vivo* (*Halliwell, 1999*; *Yin et al., 2020*). Herbicide stress, similar to other biotic and abiotic stresses, creates an imbalance in energy between those received and processed by plants (*Tuladhar, Sasidharan & Saudagar, 2021*). This inequity usually creates photo inhibition, ROS formation, and a decline in growth capacity, consequently activating or accelerating cell death (*Tripathy & Oelmüller, 2012*). Plants have developed a separate mechanism for the dissipation of these surplus amounts of energy during its due course entry in the electron transport chain component of photosynthetic machinery (*Havaux & Kloppstech, 2001*; *Asada, 1999*). It has been reported that under suboptimal conditions, the diversion of carbon flow shifted to secondary metabolism instead of primary, which leads to the synthesis of phenolic compounds. These compounds act as energy escape valves (*Hernandez & Van Breusegem, 2010*) by dissipating excess energy as fluorescence.

Henceforth, antioxidant activity enhancement under both the DPPH and FRAP assay in the present study illustrates the involvement of the enzymatic and non-enzymatic antioxidant system machinery in tandem to overcome the imazethapyr-induced stress.

## GR and GST activities in lentil

Glutathione-S-transferase, is another candidate enzyme involved in the detoxification or inactivation of numerous substrates, including xenobiotic chemicals such as herbicides by forming conjugates with glutathione. The xenobiotic and the reactive thiol group of cysteine residue of GST are connected by this process. Furthermore, conjugates are transported to the vacuoles where detoxification is completed as glutathione conjugates are hydrolyzed. Besides, it is an important role in detoxification; compartmentalization as well as chelation of the major toxic material in plants (*Anjum et al., 2015*), GSTs successively establish a proficient defense system for plants to defend the ROS-generated effects. Herbicide selectivity among the weed and crop species depends on the tolerance of the plant, which is associated with differential routes and rates of herbicide metabolism. Generally, detoxification involves three series of steps, conversion (step-1), conjugation (step-II) and deposition (step-III). The coordinated action of all these phases/steps can detoxify the herbicides with ample speed. Further, accumulation as well as the persistence of these herbicides to phytotoxic levels are limited through this concerted mechanism. The increase in GST activity in response to increasing imazethapyr dose and sampling hour in our study suggests that all these three mechanisms worked in a fine tune to detoxify/minimize the harmful effects. The greater accumulation of metolachlor in soybean than in corn is shown to be related to greater herbicide-induced GST activity in corn than in soybean (*Scarponi, Alla & Martinetti, 1992*). The different isoforms of glutathione-s-transferase *viz.*, GST (ALA), GST (CDNB), and GST (MET) get inhibited in maize treated with isoproturon, while GST (ATR) activity is unaffected, this inhibition was pronounced at a higher dose of isoproturon (*NematAlla, Hassan & El-Bastawisy, 2008*). Thus, maize is subjected to isoproturon-induced oxidative stress, and the extent of this oxidative damage increases at higher doses with increasing time. In our present study, imazethapyr induced a dose-dependent increase in the activity of GST compared well with the report of *Shivani et al. (2022)*, who reported a considerable enhancement in the GST activity in imazethapyr-treated lentil plants. *Zabalza et al. (2007)* reported a progressive increase in the GR activity following imazethapyr treatment in peas from day one of the treatment. Furthermore, the pronounced increase in GR activities at all the respective doses throughout the assay period in our study confirms the zabalza et al report in terms of the rapid impact of imazethapyr on plant and consequent response. Stimulation in GR activity is also observed with aciflurfen (*Hameed et al., 2014*) and this enhanced activity of GR prohibited both oxidation (esp. SH-containing compounds) and lipid peroxidation. A robust decline was noticed in both glutathione content and activity of GR following the acifluorfen treatment in the presence of light in cucumber disks (*Kenyon & Duke, 1985*). In general, the sensitivity of plants against herbicides appears to be reliant on numerous factors, like species in practice, the adequate reaction of the plants under a peroxidative environment (*Schmidt & Kunert, 1986*), and the metabolism of herbicides in plants (*Soares*
*et al., 2019*). The enhancement in antioxidant compound production such as glutathione and ascorbic acid exhibits the primary response against the herbicide, aciflurfen-induced peroxidation was noticed in higher plants. This enhanced level of the compounds further stimulates the GR activity with the concurrent drop in acid-soluble thiol compounds and the pace of lipid peroxidation (*Schmidt & Kunert, 1986*). *Miteva, Ivanov & Alexieva (2010)* observed that glyphosate treatment provoked an increase in both total and oxidized glutathione in pea plants and caused activation of GR in treated organs. Enhanced thiol compounds at higher imazethapyr concentrations in our study are probably the basis for stimulation in GR activity. These enhancements of the thiol compound at varying imazethapyr dose explain the abundance of substrate as well as different isoforms of GR stimulation/activation.

### Thiol content in lentil

The increased level of thiol content across the treatment over the control in the present study ranged from It 31.77% to 74.58%. Moreover, thiol content in different sampling periods increased progressively with time. Similarly, *Aly & Mohamed (2012)* also reported a similar increase in thiol content in maize against metal ion stress. Although, the primary product in sulphate assimilation is cysteine (*Finnegan & Chen, 2012*), but glutathione is reported to be the major thiol compound of the plant cell (*Smirnoff, 1993*), which provides plant defense against various stresses (*Niu & Liao, 2016*). Thiol compound can alternate between oxidized and reduced states, thus determining the redox status of cell. Plants grown under sub-optimal environmental conditions usually experience oxidative stress that leads to an elevated level of ROS (*Foyer et al., 1997*). Damage caused by ROS is prevented by various antioxidant molecules such as ascorbic acid, phenolics and thiol compounds, particularly glutathione with their direct involvement or being a substrate of enzymes such as GR and GST in the present investigation to detoxify/neutralize the imazethapyr-associated negative consequences. The role of ascorbic acid and GSH to minimize the ROS load in cell during oxidative stress in plants is well established (*Hasanuzzaman et al., 2012*). In our present study increased thiol content in response to increasing imazethapyr dose, sampling hour and their interaction advocates the major role of these thiol compounds to defend the plant against these xenobiotic classes of molecules. The synchronized actions of these antioxidants are manifested with their involvements as substrates of APX (ascorbate peroxidase) and GR in the glutathione-ascorbate cycle (*Foyer-Halliwell-Asada pathway)* in the protection of cells from ROS-induced toxicity (*Halliwell, 1999*).

## CONCLUSIONS

Lentil is a nutritious cool-season legume crop that contains a high amount of protein. The initial growth of the crop is hampered by the massive invasion of weeds. Though there are several methods to control weeds, the manna generated by herbicides is highly effective and most preferred. The current experiment aimed to determine how imazethapyr affected the metabolism of phenol and glutathione as well as the antioxidant behaviour of the lentil seedlings at various sampling hours after its application. We found that the application of

imazethapyr enhanced the phenolic acid and flavonoid content of lentil seedlings. Additionally, the antioxidant activity of the phenolic extract in both the DPPH as well as FRAP assay was also increased. In accordance with an increase in the phenolic compounds, there was a corresponding increase in mean PAL and POD activity observed, indicating the modulation of both these enzyme activities in response to herbicide treatment. The mean thiol content also increased significantly with increasing herbicide treatments, which appeared to result from an increase in the dose-dependent GR activity. Based on our findings, it can be summarized that lentil overcome the herbicide-induced oxidative stress by stimulation of PAL enzyme while detoxifying the parent molecule by stimulation of GST activity. Consequent upon increased PAL activity, phenolic acid and flavonoid content and antioxidant activity enhanced indicating the major role played by phenolics and flavonoids to overcome herbicide-induced stress.

## ACKNOWLEDGEMENTS

We are grateful to ICAR-Indian Institute of Sugarcane Research for the approval of study leave to complete the Doctoral research.

### Funding

This research was funded by Bidhan Chandra Krishi Viswavidyalaya, Nadia, WB, India and Researchers Supporting Project number (RSP2023R347), King Saud University, Riyadh, Saudi Arabia. The Ministry of Education, Youth and Sports of the Czech Republic (S grant of MSMT CR) supported the APC of this article. The funders had no role in study design, data collection and analysis, decision to publish, or preparation of the manuscript.

### Grant Disclosures

The following grant information was disclosed by the authors:
Bidhan Chandra Krishi Viswavidyalaya: RSP2023R347.
Ministry of Education, Youth and Sports of the Czech Republic: MSMT CR.

### Competing Interests

The authors declare that they have no competing interests.

### Author Contributions

- Rajeev Kumar conceived and designed the experiments, performed the experiments, analyzed the data, prepared figures and/or tables, authored or reviewed drafts of the article, and approved the final draft.
- V. Visha Kumari conceived and designed the experiments, analyzed the data, prepared figures and/or tables, authored or reviewed drafts of the article, and approved the final draft.
- Ranjit Singh Gujjar conceived and designed the experiments, prepared figures and/or tables, authored or reviewed drafts of the article, and approved the final draft.

- Mala Kumari conceived and designed the experiments, prepared figures and/or tables, authored or reviewed drafts of the article, and approved the final draft.
- Sanjay Kumar Goswami conceived and designed the experiments, authored or reviewed drafts of the article, and approved the final draft.
- Jhuma Datta conceived and designed the experiments, authored or reviewed drafts of the article, and approved the final draft.
- Srikumar Pal conceived and designed the experiments, authored or reviewed drafts of the article, and approved the final draft.
- Sudhir Kumar Jha conceived and designed the experiments, authored or reviewed drafts of the article, and approved the final draft.
- Ashok Kumar conceived and designed the experiments, authored or reviewed drafts of the article, and approved the final draft.
- Ashwini Dutt Pathak conceived and designed the experiments, authored or reviewed drafts of the article, and approved the final draft.
- Milan Skalicky performed the experiments, analyzed the data, prepared figures and/or tables, authored or reviewed drafts of the article, and approved the final draft.
- Manzer H. Siddiqui performed the experiments, analyzed the data, prepared figures and/or tables, authored or reviewed drafts of the article, and approved the final draft.
- Akbar Hossain performed the experiments, analyzed the data, prepared figures and/or tables, authored or reviewed drafts of the article, and approved the final draft.

## Data Availability

The raw data are available in the Supplemental File.

## Supplemental Information

Supplemental information for this article can be found online at http://dx.doi.org/10.7717/peerj.16370#supplemental-information.

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
