# Peer review of "Evaluating the imazethapyr herbicide mediated regulation of phenol and glutathione metabolism and antioxidant activity in lentil seedlings"

_PeerJ, doi:10.7717/peerj.16370_

## Round 0.1 · original submission · Major Revisions

Please carefully read the reviewers' suggestions about your article. If you do not accept one or more of their suggestions, give your reasons.

**Language Note:** PeerJ staff have identified that the English language needs to be improved. When you prepare your next revision, please either (i) have a colleague who is proficient in English and familiar with the subject matter review your manuscript, or (ii) contact a professional editing service to review your manuscript. PeerJ can provide language editing services - you can contact us at [email protected] for pricing (be sure to provide your manuscript number and title). – PeerJ Staff

·

Basic reporting

The manuscript is enough interesting to include for publication in this Journal. It talks about
Regulation of phenol and glutahione metabolism an atioxidant activity in lentil seedling, maybe not a novel topic, but it s the first time where the different compounds of phenolic and flavonoid contents.
The introduction clearly state the situation being investigated.

Experimental design

My suggestion, the authors should state a clear objective (aim) of the manuscript and how it can impact to know the increase of phenol and flavonoids in lentil seedlings.

The methods are accurately explained and the results are very well laid out and in a logical sequence.

Validity of the findings

The authors found that lentil overcome the herbicide-induced oxidative stress by stimulation of PAL enzyme while detoxifying the parent molecule by stimulation of GST activity. Consequent upon increased PAL activity, phenolic acid and flavonoid content and antioxidant activity enhanced that indicate the major role played by phenolics and flavonoids to overcome herbicide-induced stress.

Additional comments

The manuscript addresses a timely and important topic, the potential impacts of phenol and flavonoids. This manuscript has potential for to know better the egulation of phenol and glutahione metabolism an antioxidant activity in lentil seedling,

Reviewer 2 ·

Basic reporting

The article was written by a clear and deductible style.

Experimental design

The info about imazethapyr written in line 88 should be referred because EU was not approved registration of this herbicide.

Why did authors give references related to antioxidants, ROS, GSR, PPO, and GR between line 108 and line 120.

How applied he herbicide rates? Please give more details. Application parameters of the herbicide may change herbicidal activity.

Please check the reference year of Shanoon in line 161

Validity of the findings

In the discussion section, the authors discussed their data with other studies, but most of these studies were conducted using herbicides belonging to different mode of action. So common backgrounds of these herbicides with imazethapyr should be added to article.

Discussion section may be most problematic part of this paper because authors interpreted increasing in PAL enzyme resulted in an increase of phenolic acid and flavonoid content and antioxidant activity and, in this wise lentils overcome herbicide-induced stress. However, no data about lentil response to imazethapyr rates were found in the paper. In general, some observations should have been given in the paper including dry matter change or phytotoxicity. Many studies on ALS inhibitors revealed that symptoms of these herbicides on the plant were not seen soon after herbicide application. So, the interpretation of authors about lentils overcoming herbicide-induced stress by increasing PAL enzyme content is not approved by these data. Additional data should have been given. The discussion should be rewritten.

Additional comments

The article may be accepted after a major revision.

---

## Round 0.2 · Major Revisions

I am sending the reviewer's comments for your revisions. Please follow the reviewer's remarks and make the required improvements to your article. If you do not accept his/her requests, please give a detailed explanation about them.

Reviewer 2 ·

Basic reporting

I suggested "The info about imazethapyr written in line 88 should be referred because EU was not approved registration of this herbicide."


Your correction is not sufficient for the reviewer's request. You wrote a reference for your statement instead of improving it. In added reference, there is no specific info about the environmental safety of imazethaphyr. There is only general knowledge about AHAS inhibitors that may be found in many articles like "Acetohidroxyacid synthase (AHAS) inhibitors are worldwide used because of their broad weed control spectrum, high selectivity, low application doses and low mammalian toxicity".

Experimental design

I suggested "How applied he herbicide rates? Please give more details. Application parameters of the herbicide may change herbicidal activity."


Your correction is not sufficient for this request. Herbicides are generally applied by a sprayer. You should have provided the detail about this treatment such as which type of sprayer was used (hand type, knapsack, spray cabinet), which type of nozzle is used for the application, how much water was used her pot or what was the application?

Validity of the findings

I suggested "Many studies on ALS inhibitors revealed that symptoms of these herbicides on the plant were not seen soon after herbicide application. So, the interpretation of authors about lentils overcoming herbicide-induced stress by increasing PAL enzyme content is not approved by these data. Additional data should have been given. The discussion should be rewritten."



The discussion was not rewritten by the authors; some sentences were only rephrased.

Additional comments

Briefly, the efficacy of one herbicide on plant physiology is only valuable if this impact is associated with herbicidal impact. Changing plant enzymes following an herbicide application is important, but these results were only acceptable as a scientific report instead of a research article. I strongly suggest that the discussion must be rewritten.

---

## Round 0.3 · accepted · Accept

I would like to thank you for accepting of referees' suggestions and improving your article based on their suggestions. I think your article is ready to publish. We look forward to your next article.